# Extended Chaotic-Map-Based User Authentication and Key Agreement for HIPAA Privacy/Security Regulations

**Yi-Pei Hsieh [1], Kuo-Chang Lee [1], Tian-Fu Lee [2],\* and Guo-Jun Su [2]**

1   Department of Information Technology and Management, Tzu Chi University of Science and Technology, Hualien 97005, Taiwan; hsiehyp@tcust.edu.tw (Y.-P.H.); kclee@tcust.edu.tw (K.-C.L.)
2   Department of Medical Informatics, Tzu Chi University, Hualien 97004, Taiwan; 103325107@gms.tcu.edu.tw
\*   Correspondence: jackytflee@mail.tcu.edu.tw; Tel.: +886-3-856-5301 (ext. 2403)

**Abstract:** Background: The US government has enacted the Health Insurance Portability and Accountability Act (HIPAA), in which patient control over electronic protected health information is a major issue of concern. The two main goals of the Act are the privacy and security regulations in the HIPAA and the availability and confidentiality of electronic protected health information. The most recent authenticated key-agreement schemes for HIPAA privacy/security have been developed using time-consuming modular exponential computations or scalar multiplications on elliptic curves to provide higher security. However, these authenticated key-agreement schemes either have a heavy computational cost or suffer from authorization problems. Methods: Recent studies have demonstrated that cryptosystems using chaotic-map operations are more efficient than those that use modular exponential computations and scalar multiplications on elliptic curves. Additionally, enhanced Chebyshev polynomials exhibit the semigroup property and the commutative property. Hence, this paper develops a secure and efficient certificate-based authenticated key-agreement scheme for HIPAA privacy/security regulations by using extended chaotic maps. Results and Conclusions: This work develops a user-authentication and key-agreement scheme that solves security problems that afflict related schemes. This proposed key-agreement scheme depends on a certificate-management center to enable doctors, patients and authentication servers to realize mutual authentication through certificates and thereby reduce the number of rounds of communications that are required. The proposed scheme not only provides more security functions, but also has a lower computational cost than related schemes.

**Keywords:** chaotic maps; HIPAA; authentication; key agreement; PHI security



## 1. Introduction

The network environment is accessible to the public. Communications between a pair of parties may be wiretapped or forged. Before a communication, parties must go through a key-negotiation phase to generate a session key that will protect transmitted information. Therefore, user-authentication and key-agreement schemes are necessary. The user-authentication and key-agreement scheme is used to verify the identities of both parties and prevent an attacker from fooling a user and the server by forging identities during the key-negotiation phase. It resists potential attacks and raises its own security issues. User authentication and key agreements can also guarantee the fairness of establishment of the session key. Neither of a pair of communicating parties can decide the session key in advance of their communication. The session key must be composed of information that is provided by both parties to ensure that neither party can precalculate it, resulting in an information leak.

### 1.1. Background

The US government promulgated the HIPAA privacy/security regulations [1] in 1996 to improve overall healthcare quality. The HIPAA specification is a conceptual guide-

line that can be used to design medical-related protocols. It has become very popular because it simplifies health policies and procedures and promotes the security and privacy of patients' medical information. Recently, many fragmentary and ambiguous medical regulations have been made clearer and more complete by reference to the HIPAA specification. In the traditional medical environment, the entire course of treatment is described fully in hospital and paper medical records. Nowadays, owing to the development of the network, medical records are efficiently transmitted among hospitals. Medical staff and patients can quickly and conveniently obtain related medical services. Accordingly, the security of transmission and privacy of electronic medical records has become increasingly significant. Therefore, establishing secure communication channels between patients and healthcare centers with mutual authentication and session-key negotiation is extremely important.

*1.2. HIPAA Privacy/Security Regulations*

The HIPAA privacy/security regulations [1] are briefly summarized as follows.

### 1.2.1. Privacy Regulations

Privacy regulations give patients the right to claim medical records, including protected health-related information such as name, address, contact numbers, and medical information.

### 1.2.2. Security Regulations

1. Patients' understanding: The patient has the right to know how their health information will be used and preserved. Digital signatures can be used to protect patient health information.
2. Confidentiality: Confidentially concerns protections associated with the use of software. Patient health information must be encrypted and protected in both storage and transmission to ensure confidentiality. Encryption is the most effective way to achieve the confidentiality of information.
3. Patients' control: Patients can control access to their own information by using generated and issued encryption and decryption keys.
4. Data Integrity: The integrity of e-health information must be ensured. Medically negligent use, tampering and unauthorized destruction of patients' health information are prohibited.
5. Consent Exception: When an emergency or special circumstance arises, the disclosure of a patient's medical records and health information without the patient's authorization is permitted. When this exception is used, the patient is not directly involved, so other methods of decrypting the ciphertext must be designed.

*1.3. Threat Models*

Threat models for authentication schemes in smart mobile devices are divided into the following five main categories according to the security attributes that the attack attempts to compromise [2].

1. Identity-based attacks: This attack targets authentication and attempts to forge identities to gain access to the system posing as an authorized user.
2. Eavesdropping-based attacks: This attack targets confidentiality and is based on eavesdropping on the communication channel between the user and the server to obtain some secret information and break the confidentiality of the system.
3. Combined eavesdropping and identity-based attacks: This attack targets confidentiality and authentication, and combines eavesdropping and identity-based techniques to compromise systems.
4. Manipulation-based attacks: This attack targets data integrity and involves an unauthorized party accessing and changing sensitive data.
5. Service-based attacks: This attack targets availability and attempts to make the authentication service unavailable. After that, legitimate users cannot log in to the server.



### 1.4. Related Works

Many authentication and key-agreement schemes have been proposed for e-health systems. In 2010, Hu et al. [3] proposed an authentication scheme with contract-oriented hybrid public-key infrastructure based on the HIPAA specification design electronic medical method in [4]. In 2012, Ray and Biswas [5] pointed out shortcomings of previous schemes, including that of Hu et al., including the fact that without patient authorization, medical service providers can access patient information without restriction. They also revealed that the previous schemes in [4,6] raise the problem that a round-oriented health smartcard cannot verify the need for health information in multiple places at once, and so proposed a contract-oriented CA-based electronic health service system. In 2014, Ray and Biswas [7] developed a CA-based authentication scheme for e-healthcare systems. Their developed scheme uses the existing PKI and public-key certificate to set up a contract-based system with a medical-center server located at hospitals, and is compliant with HIPAA privacy/security regulations.

In 2019, Aghili et al. [8] proposed a lightweight authentication and ownership transfer protocol for e-health systems in the context of IoT. Their protocol not only provides authentication and key agreement but also satisfies access control and preserves the privacy of doctors and patients. In 2020, Bui et al. [9] proposed a new biometric-based key-management scheme to facilitate remote-access authorization anytime and anywhere. In their scheme, patients and doctors realize mutual authentication by using their biometric information through real-time video-communication technology. Additionally, their scheme also provided a safety channel in delivering their access authorization and secret data between patient and doctor. In the same year, Ali et al. [10] presented a robust authentication and access-control protocol for securing wireless healthcare sensor networks. Their proposed scheme employed three factors, including smart-card authentication, biometric authentication and password authentication, to overcome the pitfalls in previous schemes [11,12]. Additionally, Fotouhi et al. [13] proposed a hash-chain-based authentication scheme for wireless body-area networks in healthcare IoT. Their scheme provides perfect forward secrecy and resists potential attacks, including key-compromise impersonation attacks and known session-specific temporary information attacks. In 2021, Lee et al. [14] considered the entire process of data from data generation through transmission by wearable devices to mobile devices and then to a medical center server, and developed an efficient authentication scheme based on extended chaotic maps. Their scheme reduces the amount of computation on wearable devices, while also taking advantage of the immutability of the blockchain to ensure that data cannot be tampered with, enhancing security requirements. In 2022, Amintoosi et al. [15] performed cryptanalysis of the scheme of Aghili et al. [8] and stated that it is insecure against some possible attacks. They also proposed a lightweight authentication scheme for smart healthcare applications in IoMT as an alternative. At the same year, Zhai and Wang [16] proposed an effective multiserver biometric-authentication scheme based on extended chaotic maps for TMIS to overcome the weaknesses of Lee et al.'s scheme [17] in terms of authentication and revocation. In 2022, Ryu et al. [18] introduced a new method of high-speed symmetric encryption using the Chebyshev chaotic map and developed a multiserver/multiclient authentication scheme using this symmetric map to overcome the weaknesses of Chatterjee et al.'s scheme [19] in terms of revocation and user anonymity.

### 1.5. Motivation and Contributions

In summary, the aforementioned schemes are limited by permanent authorization, difficulty of changing the password that is kept in the smartcard, the inability of a round-oriented smartcard to verify simultaneously the need for health information in multiple places, and poor computing efficiency during medical treatment.

In order to solve these problems, this investigation proposes a user-authentication and key-agreement scheme that complies with HIPAA security regulations by using enhanced Chebyshev polynomials. Many recent studies showed some mathematical models and theo-

rems can be applied to information systems and have good computing performance [20–30]. Enhanced Chebyshev polynomials exhibit semigroup and commutative properties and provide the Logarithm problem and the Diffie–Hellman problem. Additionally, recent investigations have established that cryptosystems that are developed using extended Chebyshev chaotic maps are more efficient than those developed using modular exponentiations and scalar multiplications on elliptic curves. Therefore, the efficiency of the scheme that is proposed in this investigation is enhanced by using extended Chebyshev chaotic maps. In the mutual-authentication and key-agreement phase of each session, different authorizations are generated to be compliant with HIPAA privacy regulations and security regulations. The contributions of this work are summarized as follows:

1.  A secure and efficient authentication and key-agreement scheme that is based on extended Chebyshev chaotic maps is proposed by using lightweight extended Chebyshev chaotic maps and hash operations.
2.  The proposed scheme solves the security problems of previous schemes, which do not include updated passwords, patients' authorization and patients' control, and cannot resist password-guessing attacks, impersonation attacks, replay attacks and stolen verifier attacks.
3.  The proposed scheme is compliant with HIPAA privacy and security regulations.

### 1.6. Organization

The remainder of this paper is organized as follows: Section 2 briefly introduces primitives used in this paper. Section 3 presents the proposed extended chaotic map-based user-authentication and key-agreement scheme. Section 4 presents the authentication proof using BAN logic, analyzes the security of the proposed scheme and compares the proposed scheme with the related works. Section 5 concludes this work.

## 2. Preliminaries

This section presents the notation and definitions that are used in this paper, including those related to enhanced Chebyshev polynomials, the extended chaotic-map-based discrete logarithm problem and the extended chaotic-map-based Diffie–Hellman problem.

### 2.1. Notation

A patient is denoted as *Pat*; a doctor is denoted as *Doc* and a medical-center server is denoted as *MCS*. Table 1 lists the entire notation that is used in this paper.

**Table 1.** Notations.

| Notation | Description |
| --- | --- |
| $E(.)/D(.)$ | Symmetric en/decryption algorithm, ex. DES, AES [31] |
| $h(.)$ | One-way hash function, ex. MD5, SHA-256 [31] |
| $ID_P, ID_D$ | *Pat*'s identity/*Doc*'s identity |
| $PW_P$ | *Pat*'s password |
| $NID_P/NID_D$ | *Pat*'s anonymous information/*Doc*'s anonymous information |
| $w$ | *Pat*'s medical power of attorney |
| $r_P/T_{r_P}(x)$ | *Pat*'s private/public key pair |
| $r_D/T_{r_D}(x)$ | *Doc*'s private/public key pair |
| $r_{MCS}/T_{r_{MCS}}(x)$ | *MCS*' private/public key pair |
| $PHI$ | *Pat*' Protected Health Information |
| $p$ | A large prime number |
| $V_i$ | Confirmation message |

### 2.2. Enhanced Chebyshev Polynomials

In 2008, to avoid the limitations demonstrated by Bergamo et al. [32], Zhang [33] developed the enhanced Chebyshev polynomials, and showed that the semigroup property and the commutative under composition are still satisfied. That is,

$$T_n(x) = (2T_{n-1}(x) - T_{n-2}(x))(mod\ p), \tag{1}$$

where $n \geq 2$, $x \in (-\infty, +\infty)$ and $p$ is a large prime number. Then,

$$T_r(T_s(x)) \equiv T_{r \cdot s}(x) \equiv T_s(T_r(x)) mod\ p \tag{2}$$

holds, where $r, s \geq 2$.

The enhanced Chebyshev chaotic maps still have the discrete logarithm problem and Diffie–Hellman problem [15–17], which are described as follows.

1.  Extended Chaotic-Map-Based Discrete Logarithm Problem (ECM-DLP):

    Given $x$, $y$ and $p$, it is computationally infeasible to find an integer $r$ such that

    $$T_r(x) mod\ p = y \tag{3}$$

holds.

2.  Extended Chaotic-Map-Based Diffie–Hellman Problem (ECM-DHP):

    Given $T_r(x)(mod\ p)$, $T_s(x)(mod\ p)$, $T(\cdot)$, $x \in (-\infty, +\infty)$ and $p$ is a large prime number, it is computationally infeasible to calculate

    $$T_r(T_s(x)) \equiv T_{r \cdot s}(x) \equiv T_s(T_r(x)) mod\ p, \tag{4}$$

where $r, s \geq 2$.

### 2.3. The Medical System Model

The system model includes three roles: patient *Pat*, doctor *Doc* and medical-center server *MCS*. The medical-center server *MCS* provides *Pat* with registration and issues smartcards and is responsible for providing medical services. Doctors or medical staff *Doc* must first register with the *MCS* to obtain medical staff credentials. Patients *Pat* need to sign a privacy contract with medical-center server *MCS*, obtain a smartcard *SC* during the registration phase, and then receive medical services using the smartcard *SC*. *Pat* and *Doc* store and exchange protected medical records and health information in the cloud via the help of *MCS*. Figure 1 illustrates the relations among *Pat*, *Doc* and *MCS* in the medical system model.

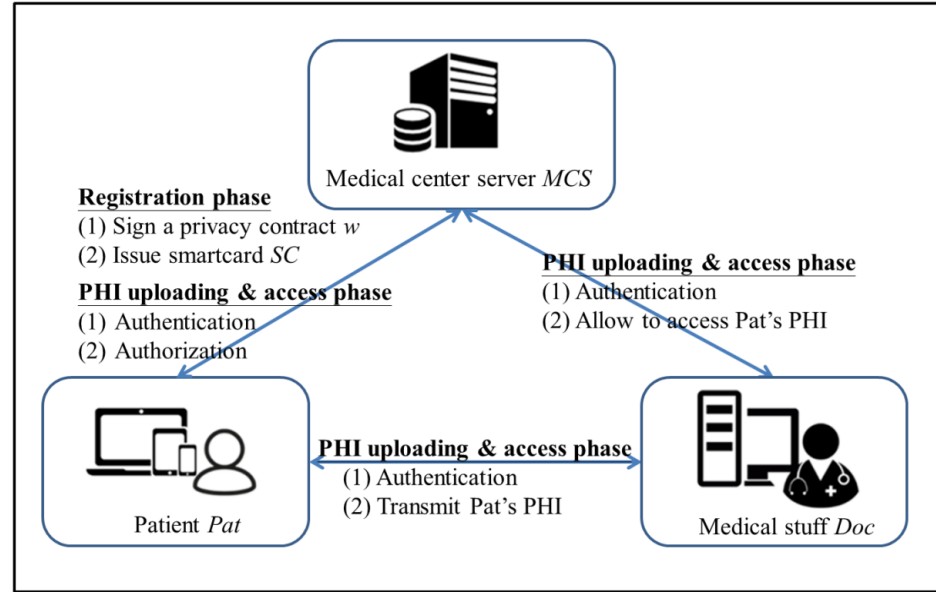

**Figure 1.** The relations between *Pat*, *Doc* and *MCS* in the medical system model. The *MCS* provides *Pat* and *Doc* with registration. *Pat* needs to sign a privacy contract with *MCS*. *Pat* and *Doc* store and exchange protected PHI via the help of *MCS*.

### 3. Proposed Extended Chaotic-Map-Based User-Authentication and Key-Agreement Scheme

This section presents a user-authentication and key-agreement scheme, which is certificate-based and used for HIPAA privacy/security regulations. The proposed scheme uses a user certificate to reduce the number of rounds of transmission and has higher security. The scheme includes three roles, which are a trusted government health server, patients and medical staff. The proposed authentication and key-agreement schemes have six phases, which are system-parameter initialization, registration, uploading of patient's *PHI*, access to patient's *PHI*, emergency-exception handling and smartcard password changing, which are described below.

*3.1. System-Parameter Initialization Phase*

Medical-center server *MCS* selects a secure hash function $h(\cdot)$, a random number $r_{MCS}$ and a random variable $x$ in $(-\infty, +\infty)$. Then, *MCS* computes $T_{r_{MCS}}(x) \bmod p$ and publishes $\{x, T_{r_{MCS}}(x) \bmod p, h(.)\}$. Patient *Pat* selects a random number $r_P$, computes $T_{r_P}(x) \bmod p$ and publishes $\{T_{r_P}(x) \bmod p\}$. Medical service staff (or Doctor) *Doc* selects a random number $r_D$, computes $T_{r_D}(x) \bmod p$ and publishes $\{T_{r_D}(x) \bmod p\}$.

*3.2. Registration Phase*

Figure 2 illustrates the processes of registration phase of the proposed scheme. Each patient *Pat* signs a privacy contract $w$ that includes the patient's information and instructions on how to be stored and used, and then performs the following steps for registration.

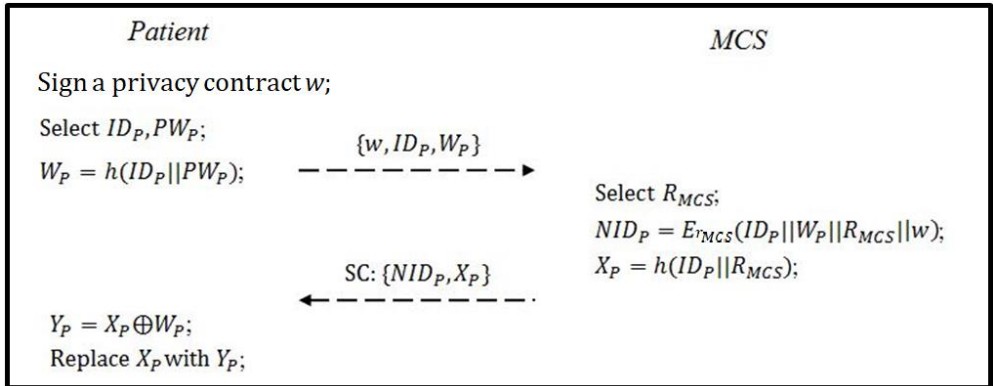

**Figure 2.** The processes of registration phase of the proposed scheme.

Step 1: Patient *Pat* selects an identity $ID_P$ and password $PW_P$, computes $W_P = h(ID_P \| PW_P)$ and sends $\{w, ID_P, W_P\}$ to *MCS*.

Step 2: On receiving $\{w, ID_P, W_P\}$ from *Pat*, MCS selects a random number $R_{MCS}$ and computes $NID_P = E_{r_{MCS}}(ID_P \| W_P \| R_{MCS} \| w)$ and $X_P = h(ID_P \| R_{MCS})$ for authentication. Then, *MCS* sends $\{NID_P, X_P\}$ to *Pat*.

Step 3: On receiving $\{NID_P, X_P\}$ from *MCS*, *Pat* obtains his/her smartcard and computes $Y_P = X_P \oplus W_P$, and replaces $X_P$ with $Y_P$.

*3.3. PHI Uploading Phase*

Figure 3 illustrates the processes of the *PHI* uploading phase of the proposed scheme, which are described as follows.

Step 1: Patient *Pat* inputs his/her identity $ID_P$ and password $PW_P$ and computes $W_P = h(ID_P \| PW_P)$; $X_P = Y_P \oplus W_P$, selects random numbers $R_P$ and $a$, a timestamp $TS_1$, computes $T_a(x) \bmod p$, $P_P = h(W_P \| ID_P \| ID_D \| T_a(x))$, $Cer_P = h(T_{r_P}(T_{r_{MCS}}(x) \bmod p) \| TS_1)$, $C_P = Cer_P \oplus R_P$, $V_1 = h(X_P \| P_P \| Cer_P \| R_P)$, $SK_{PD} = h(T_a(T_{r_D}(x) \bmod p))$, $NID_{PD} = ID_P \oplus SK_{PD}$, $Cer_{PD} = h(T_{r_P}(T_{r_D}(x) \bmod p) \| TS_1)$, $V_2 = h(ID_P \| Cer_{PD})$ and sends $\{NID_{PD}, NID_P, V_1, V_2, C_P, T_a(x), TS_1\}$ to the hospital.

**Figure 3.** The processes of PHI uploading phase of the proposed scheme.

Step 2: After receiving the messages from *Pat*, the doctor *Doc* checks the timestamp $TS_1 \leqq \Delta T$ and computes $SK_{PD} = h(T_{r_D}(T_a(x) \bmod p))$, $ID_P = NID_{PD} \oplus SK_{PD}$, $Cer_{PD} = h(T_{r_D}(T_{r_P}(x) \bmod p)\|TS_1)$ and $V_2' = h(ID_P\|Cer_{PD})$. If $V_2' = V_2$ does not hold, then *Doc* rejects this request; otherwise, *Doc* successfully authenticates *Pat*, selects random numbers $R_D$, $b$ and timestamp $TS_2$, and computes $T_b(x)$, encrypted session key $SK_{DMCS} = h(T_b(T_{r_{MCS}}(x) \bmod p))$, temporal identity $NID_D = E_{SK_{DMCS}}(ID_D\|R_D)$, $Cer_D = h(T_{r_D}(T_{r_{MCS}}(x) \bmod p)\|TS_2)$, $D_{PHI} = Cer_D \oplus PHI$ and $V_3 = h(ID_D\|Cer_D\|PHI\|R_D)$. Then, *Doc* sends $\{NID_D, NID_P, V_1, V_3, C_P, D_{PHI}, T_a(x), T_b(x), TS_1, TS_2\}$ to *MCS*.

Step 3: On receiving the messages from *Doc*, *MCS* checks the timestamp $TS_2 \leqq \Delta T$, and computes $SK_{DMCS} = h(T_{r_{MCS}}(T_b(x) \bmod p))$, $(ID_D\|R_D) = D_{SK_{DMCS}}(NID_D)$, $Cer_D = h(T_{r_{MCS}}(T_{r_D}(x) \bmod p)\|TS_2)$, $PHI = D_{PHI} \oplus Cer_D$, $V_3 = h(ID_D\|Cer_D\|PHI\|R_D)$. If $V_3' = V_3$ does not hold, then *MCS* rejects this service request; otherwise, *MCS* successfully authenticates the doctor *Doc*, computes $(ID_P\|W_P\|R_{MCS}\|w) = D_{r_{MCS}}(NID_P)$, $Cer_P = h(T_{r_P}(T_{r_{MCS}}(x) \bmod p)\|TS_1)$, $R_P = C_P \oplus Cer_P$, $P_P' = h(W_P\|ID_P\|ID_D\|T_a(x))$,

$X'_P = h(ID_P \| R_{MCS})$ and $V'_1 = h(X'_P \| P'_P \| Cer_P \| R_P)$. If $V'_1 = V_1$ holds, then *MCS* successfully authenticates the patient *Pat*, and stores *PHI* into its database. Then *MCS* chooses $R_{new}, c$, and computes $T_c(x) \bmod p$, $SK_{MCSP} = h(T_c(T_a(x) \bmod p))$, a new temporal identity $NID_P^{new} = E_{r_{MCS}}(ID_P \| W_P \| R_{new} \| w)$, $X_P^{new} = h(ID_P \| R_{new})$, $QID_P = E_{SK_{MCSP}}(NID_P^{new} \| X_P^{new})$, $V_4 = h(ID_D \| R_D)$ and $V_5 = h(NID_P^{new} \| ID_P \| SK_{MCSP} \| R_P \| X_P^{new})$. Then *MCS* sends $\{QID_P, V_4, V_5, T_c(x)\}$ to *Doc*.

Step 4: On receiving $\{QID_P, V_4, V_5, T_c(x)\}$ from *MCS*, *Doc* computes $V'_4 = h(ID_D \| R_D)$ and checks whether $V'_4 = V_4$ holds or not. If unsuccessful, *D* aborts this session. Otherwise, *Doc* sends $\{QID_P, V_5, T_c(x)\}$ to *Pat*.

Step 5: On receiving $\{QID_P, V_5, T_c(x)\}$ from *Doc*, *Pat* computes $SK_{MCSP} = h(T_a(T_c(x) \bmod p))$, decrypts $QID_P$ with $SK_{MCSP}$ and obtains $(NID_P^{new} \| X_P^{new})$. Then, *Pat* computes $V'_5 = h(NID_P^{new} \| ID_P \| SK_{MCSP} \| R_P \| X_P^{new})$ and checks whether $V'_5 = V_5$ holds or not. If unsuccessful, *Pat* aborts this request. Otherwise, *Pat* computes $Y_P^{new} = X_P^{new} \oplus W_P$ and replaces $\{NID_P, Y_P\}$ as $\{NID_P^{new}, Y_P^{new}\}$ in SC.

*3.4. PHI Access Phase*

Figure 4 illustrates the processes of the *PHI* access phase of the proposed scheme, which are described as follows.

**Figure 4.** The processes of PHI access phase of the proposed scheme.

Step 1: Patient *Pat* inputs his/her ID and password and computes $W_P = h(ID_P \| PW_P)$; $X_P = Y_P \oplus W_P$, selects $R_P$, $a$, timestamp $TS_1$, computes $T_a(x) \bmod p$, $P_P = h(W_P \| ID_P \| ID_D \| T_a(x))$, $Cer_P = h(T_{r_P}(T_{r_{MCS}}(x) \bmod p) \| TS_1)$, $C_P = Cer_P \oplus R_P$, $V_1 = h(X_P \| P_P \| Cer_P \| R_P)$, $SK_{PD} = h(T_a(T_{r_D}(x) \bmod p))$, $NID_{PD} = ID_P \oplus SK_{PD}$, and sends $\{NID_{PD}, NID_P, V_1, V_2, C_P, T_a(x), TS_1\}$ to the hospital.

Step 2: After receiving the messages from *Pat*, the doctor *Doc* checks the timestamp $TS_1 \leqq \Delta T$ and computes $SK_{PD} = h(T_{r_D}(T_a(x) \bmod p))$, $ID_P = NID_{PD} \oplus SK_{PD}$, $Cer_{PD} = h(T_{r_D}(T_{r_P}(x) \bmod p) \| TS_1)$ and $V_2' = h(ID_P \| Cer_{PD})$. If $V_2' = V_2$ does not hold, then *Doc* rejects this request; otherwise, *Doc* successfully authenticates *Pat*, selects random numbers $R_D$, $b$ and timestamp $TS_2$, and computes $T_b(x) \bmod p$, encrypted session key $SK_{DMCS} = h(T_b(T_{r_{MCS}}(x) \bmod p))$, temporal identity $NID_D = E_{SK_{DMCS}}(ID_D \| R_D \| IND_{PHI})$, $Cer_D = h(T_{r_D}(T_{r_{MCS}}(x) \bmod p) \| TS_2)$ and $V_3 = h(ID_D \| Cer_D \| R_D \| IND_{PHI})$, where $IND_{PHI}$ is the medical-record number that *Doc* requires. Then, *Doc* sends $\{NID_D, NID_P, V_1, V_3, C_P, T_a(x), T_b(x), TS_1, TS_2\}$ to *MCS*.

Step 3: On receiving the messages from *Doc*, *MCS* checks the timestamp $TS_2 \leqq \Delta T$, and computes $SK_{DMCS} = h(T_{r_{MCS}}(T_b(x) \bmod p))$ by using $r_{MCS}$, $(ID_D \| R_D \| IND_{PHI}) = D_{SK_{DMCS}}(NID_D)$, $Cer_D = h(T_{r_{MCS}}(T_{r_D}(x) \bmod p) \| TS_2)$ and $V_3 = h(ID_D \| Cer_D \| R_D \| IND_{PHI})$. If $V_3' = V_3$ does not hold, then *MCS* rejects this service request; otherwise, *MCS* successfully authenticates *Doc* and computes $(ID_P \| W_P \| R_{MCS} \| w) = D_{r_{MCS}}(NID_P)$, $Cer_P = h(T_{r_P}(T_{r_{MCS}}(x) \bmod p) \| TS_1)$, $R_P = C_P \oplus Cer_P$, $P_P' = h(W_P \| ID_P \| ID_D \| T_a(x))$, $X_P' = h(ID_P \| R_{MCS})$ and $V_1' = h(X_P' \| P_P' \| Cer_P \| R_P)$. If $V_1' = V_1$ holds, then *MCS* successfully authenticates *Pat*, and stores *PHI* in its database by using $IND_{PHI}$. Then, *MCS* chooses $R_{new}$, $c$ and computes $T_c(x)$, $SK_D = h(T_c(T_b(x) \bmod p))$, $D_{PHI} = E_{SK_D}(PHI)$, $V_4 = h(ID_D \| SK_D \| R_D \| IND_{PHI} \| h(PHI))$, $SK_{MCSP} = h(T_c(T_a(x) \bmod p))$, a new temporal identity $NID_P^{new} = E_{r_{MCS}}(ID_P \| W_P \| R_{new} \| w)$, $X_P^{new} = h(ID_P \| R_{new})$, $QID_P = E_{SK_{MCSP}}(NID_P^{new} \| X_P^{new})$ and $V_5 = h(NID_P^{new} \| ID_P \| SK_{MCSP} \| R_P \| X_P^{new})$. Then *MCS* sends $\{QID_P, V_4, V_5, D_{PHI}, T_c(x)\}$ to *Doc*.

Step 4: On receiving $\{QID_P, V_4, V_5, T_c(x)\}$ from *MCS*, *Doc* computes $SK_D = h(T_b(T_c(x) \bmod p))$, $PHI = D_{SK_D}(D_{PHI})$, $V_4' = h(ID_D \| SK_D \| R_D \| IND_{PHI} \| h(PHI))$ and checks whether $V_4' = V_4$ holds or not. If unsuccessful, D aborts this session. Otherwise, *Doc* sends $\{QID_P, V_5, T_c(x)\}$ to *Pat*.

Step 5: On receiving $\{QID_P, V_5, T_c(x)\}$ from *Doc*, *Pat* computes $SK_{MCSP} = h(T_a(T_c(x) \bmod p))$ and decrypts $QID_P$ with $SK_{MCSP}$ and obtains $(NID_P^{new} \| X_P^{new})$. Then, *Pat* computes $V_5' = h(NID_P^{new} \| ID_P \| SK_{MCSP} \| R_P \| X_P^{new})$ and checks whether $V_5' = V_5$ holds or not. If unsuccessful, *Pat* aborts this request. Otherwise, *Pat* computes $Y_P^{new} = X_P^{new} \oplus W_P$ and replaces $\{NID_P, Y_P\}$ as $\{NID_P^{new}, Y_P^{new}\}$ in SC.

*3.5. Emergency-Exception-Handling Phase*

Figure 5 illustrates the processes of the emergency-exception-handling phase of the proposed scheme, which are described as follows.

Step 1: Doctor *Doc* selects random numbers $R_D$, $a$ and timestamp $TS_1$, computes $T_a(x) \bmod p$, $Cer_D = h(T_{r_D}(T_{r_{MCS}}(x) \bmod p) \| TS_1)$, $Req_{PHI} = E_{Cer_D}(ID_P \| R_D \| IND_{PHI})$, where $IND_{PHI}$ is the medical-record index number that *Doc* requires. $V_1 = h(ID_D \| ID_P \| Cer_D \| T_a(x) \| R_D)$. Then, *Doc* sends $\{ID_D, Req_{PHI}, V_1, T_a(x), TS_1\}$ to *MCS*.

Step 2: On receiving the messages from *Doc*, *MCS* checks the timestamp $TS_1 \leqq \Delta T$, and computes $SK_{DMCS} = h(T_{r_{MCS}}(T_b(x) \bmod p))$ by using $r_{MCS}$, $(ID_D \| R_D \| IND_{PHI}) = D_{SK_{DMCS}}(Req_{PHI})$, $Cer_D = h(T_{r_{MCS}}(T_{r_D}(x) \bmod p) \| TS_1)$, $V_1 = h(ID_D \| ID_P \| Cer_D \| T_a(x) \| R_D)$. If $V_1' = V_1$ does not hold, then *MCS* rejects this service request; otherwise, *MCS* successfully authenticates *Doc*, stores $PHI_P$ in its database by using the index $IND_{PHI}$, selects $TS_2$, $b$ and computes $T_b(x) \bmod p$, $SK_D = h(T_b(T_a(x) \bmod p))$, $D_{PHI} = E_{SK_D}(PHI_P)$, $V_2 = h(ID_D \| SK_D \| h(PHI_P) \| TS_2 \| R_D)$. Then, *MCS* sends $\{D_{PHI}, V_2, T_b(x), TS_2\}$ to *Doc*.

$$Doc\ (r_D, T_{r_D}(x))$$

$$MCS\ (r_{MCS}, T_{r_{MCS}}(x))$$

Select $R_D, TS_1, a$;
Compute $T_a(x)$;
$Cer_D = h(T_{r_D}(T_{r_{MCS}}(x))||TS_1)$;
$Req_{PHI} = E_{Cer_D}(ID_P||R_D||IND_{PHI})$;
$V_1 = h(ID_D||ID_P||Cer_D||T_a(x)||R_D)$;

Check $TS_1 \leqq \Delta T$;
$Cer_D = h(T_{r_{MCS}}(T_{r_D}(x))||TS_1)$;
$(ID_P||R_D||IND_{PHI}) = D_{Cer_D}(Req_{PHI})$;
$V_1 = h(ID_D||ID_P||Cer_D||T_a(x)||R_D)$;
Check $V_1' =? V_1$;

(1) $\{ID_D, Req_{PHI}, V_1, T_a(x), TS_1\}$
$\xrightarrow{\hspace{3cm}}$

Gets $PHI_P$ by using $IND_{PHI}$;
Select $TS_2, b$;
Compute $T_b(x)$;
$SK_D = h(T_b(T_a(x)))$;
$D_{PHI} = E_{SK_D}(PHI_P)$;
$V_2 = h(ID_D||SK_D||h(PHI_P)||TS_2||R_D)$;

Check $TS_2 \leqq \Delta T$;
$SK_D = h(T_a(T_b(x)))$;
$PHI_P = D_{SK_D}(D_{PHI})$;
$V_2 = h(ID_D||SK_D||h(PHI_P)||TS_2||R_D)$;

(2) $\{D_{PHI}, V_2, T_b(x), TS_2\}$
$\xleftarrow{\hspace{3cm}}$

**Figure 5.** The processes of emergency-exception handling phase of the proposed scheme.

Step 3: On receiving $\{D_{PHI}, V_2, T_b(x), TS_2\}$ from $MCS$, $Doc$ checks the timestamp $TS_2 \leqq \Delta T$, computes $SK_D = h(T_a(T_b(x)\ mod\ p))$, $PHI_P = D_{SK_D}(D_{PHI})$, $V_2' = h(ID_D||SK_D|| h(PHI_P)||TS_2||R_D)$ and checks whether $V_2' = V_2$ holds or not. If unsuccessful, $Doc$ aborts this session. Otherwise, $Doc$ successfully authenticates $MCS$ and obtains the correct $PHI_P$ of *Pat*.

*3.6. Password-Updating Phase*

Figure 6 illustrates the processes of the password-updating phase of the proposed scheme, which are described as follows.

$$Patient\ (r_P, T_{r_P}(x))$$

$$MCS\ (r_{MCS}, T_{r_{MCS}}(x))$$

Input $ID_P, PW_P, PW_P^{new}$;
$W_P = h(ID_P||PW_P)$;
$W_P^{new} = h(ID_P||PW_P^{new})$;
$X_P = Y_P \oplus W_P$;
Select $R_P, TS_1, a,,$;
Compute $T_a(x)$;
$Cer_P = h(T_{r_P}(T_{r_{MCS}}(x))||TS_1)$;
$NPW_P = Cer_P \oplus (W_P^{new}||R_P)$;
$V_1 = h(X_P||Cer_P||R_P||T_a(x)||W_P||W_P^{new})$;

Check $TS_1 \leqq \Delta T$;
$(ID_P||W_P||R_{MCS}||w) = D_{R_{MCS}}(NID_P)$;
$Cer_P' = h(T_{r_{MCS}}(T_{r_P}(x)\ mod\ p)||TS_1)$;
$(W_P^{new}||R_P) = NPW_P \oplus Cer_P'$;
$X_P' = h(ID_P||R_{MCS})$;
$V_1 = h(X_P'||Cer_P'||R_P||T_a(x)||W_P||W_P^{new})$;
Select $R_{new}, c$;
Compute $T_c(x)$;
$SK_{MCSP} = h(T_c(T_a(x)))$;
$NID_P^{new} = E_{T_{MCS}(X)}(ID_P||W_P^{new}||R_{new}||w)$;
$X_P^{new} = h(ID_P||R_{new})$;
$QID_P = E_{SK_{MCSP}}(NID_P^{new}||X_P^{new})$;
$V_2 = h(NID_P^{new}||SK_{MCSP}||R_P||X_P^{new})$;

(1) $\{NID_P, NPW_P, V_1, T_a(x), TS_1\}$
$\xrightarrow{\hspace{3cm}}$

$SK_{MCSP} = h(T_a(T_c(x)))$;
$(NID_P^{new}||QX_P) = D_{SK_{MCSP}}(QID_P)$;
$V_2' = h(NID_P^{new}||SK_{MCSP}||R_P||X_P^{new})$;
Check $V_2' =? V_2$;
$Y_P^{new} = X_P^{new} \oplus W_P^{new}$;
Replace $NID_P, Y_P$ with $NID_P^{new}, Y_P^{new}$.

(2)$\{QID_P, V_2, T_c(x)\}$
$\xleftarrow{\hspace{3cm}}$

**Figure 6.** The processes of password-updating phase of the proposed scheme.

Step 1: Patient *Pat* inputs his/her identity $ID_P$, password $PW_P$ and a new password $PW_P^{new}$ and computes $W_P = ID_P \oplus PW_P$, $X_P = Y_P \oplus W_P$, $W_P^{new} = ID_P \oplus PW_P^{new}$.

Then, *Pat* selects random numbers $R_P$ and a, a timestamp $TS_1$, computes $T_a(x) \bmod p$, $P_P = h(W_P \| ID_P \| ID_D \| T_a(x))$, $Cer_P = h(T_{r_P}(T_{r_{MCS}}(x) \bmod p) \| TS_1)$, $NPW_P = Cer_P \oplus (W_P^{new} \| R_P)$, $V_1 = h(X_P \| Cer_P \| R_P \| T_a(x) \| W_P \| W_P^{new})$ and sends $\{NID_P, NPW_P, V_1, T_a(x), TS_1\}$ to *MCS*.

Step 2: On receiving the messages from *Pat*, *MCS* checks the timestamp $TS_1 \leqq \Delta T$, and computes $(ID_P \| W_P \| R_{MCS} \| w) = D_{r_{MCS}}(NID_P)$, $Cer'_P = h(T_{r_P}(T_{r_{MCS}}(x) \bmod p) \| TS_1)$, $(W_P^{new} \| R_P) = NPW_P \oplus Cer'_P$, $X'_P = h(ID_P \| R_{MCS})$ and $V_1 = h(X'_P \| Cer'_P \| R_P \| T_a(x) \| W_P \| W_P^{new})$. If $V_1 = V'_1$ holds, then *MCS* successfully authenticates the patient *Pat*, and selects $R_{new}, c$, computes $T_c(x) \bmod p$, $SK_{MCSP} = h(T_c(T_a(x) \bmod p))$, $NID_P^{new} = E_{T_{MCS}(X)}(ID_P \| W_P^{new} \| R_{new} \| w)$, $X_P^{new} = h(ID_P \| R_{new})$ and sends $\{QID_P, V_2, T_c(x)\}$ to *Pat*.

Step 3: On receiving $\{QID_P, V_2, T_c(x)\}$ from *MCS*, *Pat* computes $SK_{MCSP} = h(T_a(T_c t(x) \bmod p))$, decrypts $QID_P$ with $SK_{MCSP}$ and obtains $(NID_P^{new} \| X_P^{new})$. Then, *Pat* computes $V'_2 = h(NID_P^{new} \| SK_{MCSP} \| R_P \| X_P^{new})$ and checks whether $V'_2 = V_2$ holds or not. If unsuccessful, *Pat* aborts this request. Otherwise, *Pat* computes $Y_P^{new} = X_P^{new} \oplus W_P$ and replaces $\{NID_P, Y_P\}$ as $\{NID_P^{new}, Y_P^{new}\}$ in SC.

## 4. Security and Performance Analyses

This section presents the authentication proof using BAN logic [34], analyzes the security of the proposed scheme and provides performance and functionality comparisons between the related schemes and the proposed scheme.

### 4.1. Authentication Proof of the Proposed Scheme Using BAN Logic

This subsection shows that the proposed scheme satisfies the session-key security and mutual authentication by using BAN logic [34]. Table 2 lists the notations of BAN logic.

**Table 2.** BAN logic notations [34].

| Notation | Abbreviation |
|---|---|
| $P \mid\equiv X$ | The entity $P$ believes the statement $X$ |
| $P \Longrightarrow X$ | $P$ has jurisdiction over the statement $X$ |
| $P \mid\sim X$ | $P$ once said $X$ |
| $P \triangleleft X$ | $P$ sees $X$ |
| $\langle X \rangle_K$ | Formula $X$ is encrypted under the key $K$ |
| $P \overset{K}{\leftrightarrow} Q$ | $P$ and $Q$ communicate via shared key $K$ |
| $P \rightarrow Q : m$ | $P$ sends the message $m$ and $Q$ receives it |
| #$X$ | The message #$X$ is freshly generated |

4.1.1. Inference Rules of BAN Logic

- **Rule 1.** $\frac{P \mid\equiv P \overset{K}{\leftrightarrow} Q,\ P \triangleleft \langle X \rangle_K}{P \mid\equiv Q \sim X}$: If the entity $P$ believes that the secret $K$ is shared with $Q$ and sees message X is encrypted using $K$, then P believes that $Q$ once said $X$.

- **Rule 2.** $\frac{P \mid\equiv \#(X),\ P \mid\equiv Q \sim X}{P \mid\equiv Q \mid\equiv X}$: If the entity $P$ believes that $X$ is fresh and the entity $Q$ once said $X$, then $P$ believes that $Q$ believes $X$.

- **Rule 3.** $\frac{P \mid\equiv Q \Longrightarrow X,\ P \mid\equiv Q \mid\equiv X}{P \mid\equiv X}$: If the entity $P$ believes that $Q$ has jurisdiction over $X$ and $Q$ believes $X$, then P believes that $X$ is true.

- **Rule 4.** $\frac{P \mid\equiv \#(X),\ P \mid\equiv Q \mid\equiv X}{P \mid\equiv P \overset{K}{\leftrightarrow} Q}$: If the entity $P$ believes that $X$ is fresh and $Q$ believes $X$, then $P$ believes the secret $K$ that is shared between both entities $P$ and $Q$.

- **Rule 5.** $\frac{P \mid\equiv \#(X)}{P \mid\equiv \#(X, Y)}$: If the entity $P$ believes that $X$ is fresh, then $P$ believes the freshness of $(X, Y)$.

### 4.1.2. Goals of Authentication Proof

- **Goal 1**: $MCS \mid\equiv Pat \mid\equiv MCS \overset{SK}{\leftrightarrow} Pat$
- **Goal 2**: $MCS \mid\equiv Doc \mid\equiv MCS \overset{SK}{\leftrightarrow} Doc$
- **Goal 3:** $Doc \mid\equiv MCS \mid\equiv Doc \overset{SK}{\leftrightarrow} MCS$
- **Goal 4:** $Pat \mid\equiv MCS \mid\equiv Pat \overset{SK}{\leftrightarrow} MCS$
- **Goal 5:** $Doc \mid\equiv Pat \mid\equiv Doc \overset{SK}{\leftrightarrow} Pat$
- **Goal 6:** $Pat \mid\equiv Doc \mid\equiv Pat \overset{SK}{\leftrightarrow} Doc$

### 4.1.3. Idealized Form

- **M$_1$**. $(Pat arrow Doc) : \{NID_{PD}, NID_P, V_1 : h(X_P\|P_P\|Cer_P\|R_P), V_2 : h(ID_P\|Cer_{PD}), C_P : \langle P_P \rangle_{Cer_P}, T_a(x), TS_1\}$
- **M$_2$**. $(Doc arrow MCS) : \{NID_D, NID_P, V_1, V_3 : h(ID_D\|Cer_D\|PHI\|R_D), C_P, D_{PHI} : \langle PHI \rangle_{Cer_D}, T_a(x), T_b(x), TS_1, TS_2\}$
- **M$_3$**. $(.MCS arrow Doc) : \{QID_P : \langle NID_P^{new}\|X_P^{new}\rangle_{SK_{MCSP}}, V_4 : h(ID_D\|R_D), V_5 : h(NID_P^{new}\|ID_P\|SK_{MCSP}\|R_P\|X_P^{new}), T_c(x)\}$
- **M$_4$**. $(.Doc arrow Pat) : \{QID_P, V_5, T_c(x)\}$

### 4.1.4. Assumptions

- $AS_1$: $MCS \mid\equiv \# h(X_P\|P_P\|Cer_P\|R_P)$
- $AS_2$: $MCS \mid\equiv \# h(ID_D\|Cer_D\|PHI\|R_D)$
- $AS_3$: $Pat \mid\equiv Pat \overset{T_{r_{P}r_{MCS}}(x)}{\leftrightarrow} MCS$
- $AS_4$: $Doc \mid\equiv Doc \overset{T_{r_{D}r_{MCS}}(x)}{\leftrightarrow} MCS$
- $AS_5$: $MCS \mid\equiv MCS \overset{T_{r_{P}r_{MCS}}(x)}{\leftrightarrow} Pat$
- $AS_5$: $MCS \mid\equiv MCS \overset{T_{r_{D}r_{MCS}}(x)}{\leftrightarrow} Doc$
- $AS_7$: $MCS \mid\equiv Pat \implies R_P$
- $AS_8$: $MCS \mid\equiv Doc \implies R_D$
- $AS_9$: $Doc \mid\equiv MCS \implies h(ID_D\|R_D)$
- $AS_{10}$: $Pat \mid\equiv MCS \implies h(NID_P^{new}\|ID_P\|SK_{MCSP}\|R_P\|X_P^{new})$
- $AS_{11}$: $Doc \mid\equiv Doc \overset{T_{r_{P}r_D}(x)}{\leftrightarrow} Pat$
- $AS_{12}$: $Pat \mid\equiv Pat \overset{T_{r_{P}r_D}(x)}{\leftrightarrow} Doc$
- $AS_{13}$: $Doc \mid\equiv Pat \implies Cer_{PD}$
- $AS_{14}$: $Pat \mid\equiv Doc \implies ID_P$

### 4.1.5. Verification

By using Message **M$_2$**,

$$MCS \triangleleft \{NID_D, NID_P, V_1 : h(X_P\|P_P\|Cer_P\|R_P), \\ V_3 : h(ID_D\|Cer_D\|PHI\|R_D), C_P, D_{PHI} : PHI_{Cer_D}, T_a(x), T_b(x), TS_1, TS_2\}. \tag{5}$$

From Rule 1 and $AS_5$,
$$S_1 : MCS \mid\equiv Pat \mid\sim R_P. \tag{6}$$

From Rule 2 and $AS_1$,
$$S_2 : MCS \mid\equiv Pat \mid\equiv R_P. \tag{7}$$

From Rule 3 and $AS_7$,
$$S_3 : MCS \mid\equiv R_P. \tag{8}$$

From Rule 4, $AS_1$ and $S_2$,
$$S_4 : MCS \mid\equiv MCS \overset{SK}{\leftrightarrow} Pat.$$

Further, using Rule 2, $AS_1$ and $S_1$,

$$S_5: \ MCS \mid\equiv Pat \mid\equiv MCS \overset{SK}{\leftrightarrow} Pat. \ \textbf{Goal 1} \tag{9}$$

By using similar arguments, from Rule 1 and $AS_6$,

$$S_6: \ MCS \mid\equiv Doc \mid\sim R_D. \tag{10}$$

From Rule 2 and $AS_2$ and $S_6$,

$$S_7: \ MCS \mid\equiv Doc \mid\equiv R_D. \tag{11}$$

From Rule 3 and $AS_8$,

$$S_8: \ MCS \mid\equiv R_D. \tag{12}$$

According to Rule 4, $AS_2$ and $S_7$,

$$S_9: \ MCS \mid\equiv MCS \overset{SK}{\leftrightarrow} Doc. \tag{13}$$

Using Rule 2, $AS_2$ and $S_6$, we have

$$S_{10}: \ MCS \mid\equiv Doc \mid\equiv MCS \overset{SK}{\leftrightarrow} Doc. \ \textbf{Goal 2}$$

By using Message $\mathbf{M_3}$,

$$Doc \lhd \left\{ QID_P : \left\langle NID_P^{new} \| X_P^{new} \right\rangle_{SK_{MCSP}}, V_4 : h(ID_D \| R_D), \\ V_5 : h\left(NID_P^{new} \| ID_P \| SK_{MCSP} \| R_P \| X_P^{new}\right), T_c(x) \right\} \tag{14}$$

From Rule 1 and $AS_4$,

$$S_{11}: \ Doc \mid\equiv MCS \mid\sim \ h(ID_D \| R_D). \tag{15}$$

From Rule 5 and $(Doc \mid\equiv \# R_D)$,

$$S_{12}: \ Doc \mid\equiv \ \# h(ID_D \| R_D). \tag{16}$$

From Rule 2, $S_{11}$ and $S_{12}$,

$$S_{13}: \ Doc \mid\equiv MCS \mid\equiv h(ID_D \| R_D). \tag{17}$$

Then, from Rule 3 and $AS_9$,

$$S_{14}: \ Doc \mid\equiv h(ID_D \| R_D). \tag{18}$$

According to Rule 4, $S_{12}$ and $S_{13}$,

$$S_{15}: \ Doc \mid\equiv Doc \overset{SK}{\leftrightarrow} MCS. \tag{19}$$

Further, using Rule 2, $S_{11}$ and $S_{12}$,

$$S_{16}: \ Doc \mid\equiv MCS \mid\equiv Doc \overset{SK}{\leftrightarrow} MCS. \ \textbf{Goal 3}$$

By using similar arguments and Message $\mathbf{M_4}$,

$$Pat \lhd \{ QID_P, V_5 : h(NID_P^{new} \| ID_P \| SK_{MCSP} \| R_P \| X_P^{new}), T_c(x) \}. \tag{20}$$

From Rule 1 and $AS_3$,

$$S_{17}: \ Pat \mid\equiv MCS \mid\sim h(NID_P^{new} \| ID_P \| SK_{MCSP} \| R_P \| X_P^{new}). \tag{21}$$

From Rule 5 and $(Pat \mid\equiv \# R_D)$,

$$S_{18}: \ Pat \mid\equiv \ \# h(NID_P^{new} \| ID_P \| SK_{MCSP} \| R_P \| X_P^{new}). \tag{22}$$

From Rule 2, $V_{17}$ and $V_{18}$,

$$S_{19}: \ Pat \mid\equiv MCS \mid\equiv h(NID_P^{new} \| ID_P \| SK_{MCSP} \| R_P \| X_P^{new}). \tag{23}$$

Then, from Rule 3 and $AS_{10}$,

$$S_{20}: \ Pat \mid\equiv h(NID_P^{new}\|ID_P\|SK_{MCSP}\|R_P\|X_P^{new}). \tag{24}$$

According to Rule 4, $S_{18}$ and $S_{19}$,

$$S_{21}: \ Pat \Big|\equiv Pat \overset{SK}{\leftrightarrow} MCS. \tag{25}$$

Further, using Rule 2, $S_{17}$ and $S_{18}$,

$$S_{22}: \ Pat \Big|\equiv MCS \Big|\equiv Pat \overset{SK}{\leftrightarrow} MCS. \ \textbf{Goal 4} \tag{26}$$

By using Message **M₁,**

$$Doc \lhd \{NID_{PD}, NID_P, V_1 : h(X_P\|P_P\|Cer_P\|R_P), V_2 : h(ID_P\|Cer_{PD}), C_P : \langle P_P \rangle_{Cer_P}, T_a(x), TS_1\} \tag{27}$$

From Rule 1 and $AS_{11}$,

$$S_{23}: \ Doc \mid\equiv Pat \mid\sim Cer_{PD}. \tag{28}$$

From Rule 5 and $(Doc \mid\equiv \# TS_1)$,

$$S_{24}: \ Doc \mid\equiv \# Cer_{PD}. \tag{29}$$

From Rule 2, $V_{23}$ and $V_{24}$,

$$S_{25}: \ Doc \mid\equiv Pat \mid\equiv Cer_{PD}. \tag{30}$$

Then, from Rule 3 and $AS_{13}$,

$$S_{26}: \ Doc \mid\equiv Cer_{PD}. \tag{31}$$

According to Rule 4, $S_{24}$ and $S_{25}$,

$$S_{27}: \ Doc \Big|\equiv Doc \overset{SK}{\leftrightarrow} Pat. \tag{32}$$

Further, using Rule 2, $S_{23}$ and $S_{24}$,

$$S_{28}: \ Doc\Big|\equiv Pat \Big|\equiv Doc \overset{SK}{\leftrightarrow} Pat. \ \textbf{Goal 5} \tag{33}$$

By using Message **M₄,**

$$Pat \lhd \{QID_P, V_5 : h(NID_P^{new}\|ID_P\|SK_{MCSP}\|R_P\|X_P^{new}), T_c(x)\}. \tag{34}$$

From Rule 1 and $AS_{12}$,

$$S_{29}: \ Pat \mid\equiv Doc \mid\sim ID_P. \tag{35}$$

From Rule 2 and $S_{18}$: $Pat \mid\equiv \# h(NID_P^{new}\|ID_P\|SK_{MCSP}\|R_P\|X_P^{new})$,

$$S_{30}: \ Pat \mid\equiv Doc \mid\equiv ID_P. \tag{36}$$

Then, from Rule 3 and $AS_{14}$,

$$S_{31}: \ Pat \mid\equiv ID_P. \tag{37}$$

According to Rule 4, $S_{18}$ and $S_{30}$,

$$S_{32}: \ Pat \Big|\equiv Pat \overset{SK}{\leftrightarrow} Doc. \tag{38}$$

Further, using Rule 2, $S_{29}$ and $S_{30}$,

$$S_{33}: \ Pat \Big|\equiv Doc \Big|\equiv Pat \overset{SK}{\leftrightarrow} Doc. \ \textbf{Goal 6} \tag{39}$$

The proof is completed.

*4.2. Security Analyses*

4.2.1. Mutual Authentication (Threat Model 1)

In the proposed scheme, Medical-Center Server MCS authenticates Doctor *Doc* by checking $V_3$ since only *Doc* and *MCS* have capability to compute $Cer_D$, and thus $V_3$, where $Cer_D = h(T_{r_{MCS}r_D}(x)\bmod p\|TS_2)$, $PHI = D_{PHI} \oplus Cer_D$ and $V_3 = h(ID_D\|Cer_D\|PHI\|R_D)$. Similarly, *MCS* authenticates Patient *P* by checking $V_1$ since only *Pat* and *MCS* have

capability to compute $Cer_P$, and thus $V_1$, where $Cer_P = h(T_{r_{MCS}r_P}(x)\bmod p\|TS_1)$ and $V_1 = h(X_P\|P_P\|Cer_P\|R_P)$. Patient *Pat* authenticates *MCS* by checking $V_5$, since only *MCS* can compute $Cer_P$, and thus $R_P$, where $R_P = C_P \oplus Cer_P$ and $V_5 = h(NID_P^{new}\|ID_P\|SK_{MCSP}\|R_P\|X_P^{new})$. Patient *Pat* implicitly authenticates *Doc* through *MCS* by checking $V_1$ and $V_3$ to make sure that the doctor *Doc* is authorized by using $P_P$, where $P_P = h(W_P\|ID_P\|ID_D\|T_a(x))$. Doctor *Doc* authenticates *MCS* by checking $V_4$, since only *MCS* can compute $SK_{DMCS}$ and derive $R_D$ from $NID_D$, where $SK_{DMCS} = h(T_{br_{MCS}}(x)\bmod p)\ NID_D = E_{SK_{DMCS}}(ID_D\|R_D)$, and $V_4 = h(ID_D\|R_D)$. Doctor *Doc* implicitly authenticates *Pat* through *MCS* by checking $V_4$ since *MCS* will send $V_4$ after $P$ passes the authentication of *MCS*.

### 4.2.2. Session-Key Security (AKE-Security, Threat Model 1)

In order to ensure the security of *PHI*, the proposed scheme uses the chaotic-map-based Diffie–Hellman key exchange to negotiate the session keys $SK_{MCSP} = h(T_{a\cdot c}(x)\bmod p)$ of *Pat* and *MCS*, $SK_D = h(T_{a\cdot b}(x)\bmod p)$ of *Doc* and *MCS*, and $SK_{PD} = h(T_{a\cdot r_D}(x)\bmod p)$ of *Pat* and *Doc*. Therefore, the session-key security of the proposed scheme is based on ECM-DHP and one-way property of the hash function, and thus is negligible. Hence, the proposed scheme provides session-key security.

### 4.2.3. Resisting Password-Guessing Attacks (Threat Model 2)

1. Undetectable online password-guessing attack:

A malicious attacker *A* who has $SC : \{NID_P, Y_P\}$ can guess a password $PW_P'$, and compute $W_P' = h(ID_P'\|PW_P')$, $X_P' = Y_P \oplus W_P'$ and $P_P' = h(W_P'\|ID_P'\|ID_D\|T_a(x))$. *A* cannot compute $Cer_P$ without $r_P$ and $r_{MCS}$ because of ECM-DLP. Thus, *A* cannot successfully compute a $V_1' = h(X_P'\|P_P'\|Cer_P\|R_P)$ and send it to *MCS*. Thus, an incorrect online guess will be detected by *MCS*. Therefore, the proposed scheme resists undetectable online password-guessing attacks.

2. Offline password-guessing attack:

A malicious attacker *A* who has $SC : \{NID_P, Y_P\}$ can guess a password $PW_P'$, and compute $W_P' = h(ID_P'\|PW_P')$, $X_P' = Y_P \oplus W_P'$ and $P_P' = h(W_P'\|ID_P'\|ID_D\|T_a(x))$. *A* cannot compute $Cer_P$ without $r_P$ and $r_{MCS}$ because of ECM-DLP. Consequently, *A* cannot compute a $V_1' = h(X_P'\|P_P'\|Cer_P\|R_P)$ to verify $V_1' =?V_1$. Additionally, $W_P$, which contains $PW_P$, in $NID_P$ is encrypted with *MCS*'s secret key $r_{MCS}$, where $W_P = h(ID_P\|PW_P)$ and $NID_P = E_{r_{MCS}}(ID_P\|W_P\|R_{MCS}\|w)$. Thus, *A* cannot obtain any information about $W_P$ and $PW_P$ without $r_{MCS}$. Therefore, the proposed scheme resists offline password-guessing attacks.

### 4.2.4. Resisting Impersonation Attacks (Threat Model 1)

A malicious attacker *Pat** has $SC : \{NID_P, Y_P\}$ and tries to impersonate *Pat*. *Pat** cannot derive $W_P$ from $NID_P$, where $NID_P = E_{r_{MCS}}(ID_P\|W_P\|R_{MCS}\|w)$, because of the security of the symmetric en/decryption algorithm. Moreover, *Pat** cannot derive the private key $r_P$ from a previous message $V_1$, where $V_1 = h(X_P\|P_P\|Cer_P\|R_P)$ and $Cer_P = h(T_{r_Pr_{MCS}}(x)\bmod p\|TS_1)$, because of the one-way hash property and ECM-DLP. Hence, *Pat** cannot compute the correct $W_P$, $Cer_P$, $P_P$ and $V_1$ without the correct $PW_P$ and $r_P$, where $W_P = h(ID_P\|PW_P)$, $P_P = h(W_P\|ID_P\|ID_D\|T_a(x))$, $Cer_P = h(T_{r_P}(T_{r_{MCS}}(x)\bmod p)\|TS_1)$, $V_1 = h(X_P\|P_P\|Cer_P\|R_P)$, and cannot send out the correct $\{NID_{PD}, NID_P, V_1, V_2, C_P, T_a(x), TS_1\}$. Therefore, a login will be detected by *Doc* or *MCS*.

Similarly, a malicious attacker *Doc** tries to impersonate *Doc*. *Doc** cannot derive the private key $r_D$ from previous messages $V_2$ and $V_3$, where $Cer_{PD} = h(T_P(T_{r_D}(x)\bmod p)\|TS_1)$, $V_2 = h(ID_P\|Cer_{PD})$, $V_3 = h(ID_D\|Cer_D\|PHI\|R_D))$ and $Cer_P = h(T_{r_Pr_{MCS}}(x)\bmod p\|TS_1)$, because of the one-way hash property and ECM-DLP. Then, *Doc** cannot compute $SK_{PD} = h(T_{r_D}(T_a(x)\bmod p))$, $Cer_D = h(T_{r_D}(T_{r_{MCS}}(x)\bmod p)\|TS_2)$, $D_{PHI} = Cer_D \oplus PHI$, and $V_3 = h(ID_D\|Cer_D\|PHI\|R_D)$ without $r_D$, and so cannot send out the correct $\{NID_D, NID_P, V_1, V_3, C_P, D_{PHI}, T_a(x), T_b(x), TS_1, TS_2\}$. Therefore, a login failure will be detected by *MCS*.

### 4.2.5. Resisting Replay Attacks (Threat Model 1)

All communication messages among *Pat*, *Doc* and *MCS* contain timestamps $TS_i$ to guarantee message freshness so the proposed scheme is secure against replay attacks.

### 4.2.6. Resisting Man-in-the-Middle Attacks (Threat Model 4)

An attacker *A* intercepts patient transmission $M_1 = \{NID_{PD}, NID_P, V_1, V_2, C_P, T_a(x), TS_1\}$ and $M_4 = \{QID_P, V_5, T_c(x)\}$ between *Pat* and *Doc*. *A* cannot successfully modify $M_1$ without $PW_P$, $r_P$ or $r_D$ because of the one-way hash property and ECM-DLP, where $W_P = h(ID_P \| PW_P)$, $P_P = h(W_P \| ID_P \| ID_D \| T_a(x))$, $Cer_P = h(T_{r_P}(T_{r_{MCS}}(x) \bmod p) \| TS_1)$, $V_1 = h(X_P \| P_P \| Cer_P \| R_P)$ and $V_2 = h(ID_P \| Cer_{PD})$. *A* cannot successfully modify $M_4$ without $r_P$ and $r_{MCS}$ because of the one-way hash property and ECM-DLP, where $QID_P = E_{SK_{MCSP}}(NID_P^{new} \| X_P^{new})$, $NID_P^{new} = E_{r_{MCS}}(ID_P \| W_P \| R_{new} \| w)$, $X_P^{new} = h(ID_P \| R_{new})$, $QID_P = E_{SK_{MCSP}}(NID_P^{new} \| X_P^{new})$, and $V_5 = h(NID_P^{new} \| ID_P \| SK_{MCSP} \| R_P \| X_P^{new})$. An attacker *A* intercepts patient transmission $M_2 = \{NID_P, NID_D, V_1, V_3, C_P, D_{PHI}, T_a(x), T_b(x), TS_1, TS_2\}$ and $M_3 = \{QID_P, V_4, V_5, T_c(x)\}$ between *Doc* and *MCS*. *A* cannot successfully modify $(M_2 - M_1)$ and $(M_3 - M_4)$ without $r_{MCS}$ or $r_D$ because of the one-way hash property and ECM-DLP, where $SK_{PD} = h(T_{r_D \cdot a}(x) \bmod p)$, $ID_P = NID_{PD} \oplus SK_{PD}$, $Cer_{PD} = h(T_{r_D \cdot r_P}(x) \bmod p \| TS_1)$, $SK_{DMCS} = h(T_{b \cdot r_{MCS}}(x) \bmod p)$, $NID_D = E_{SK_{DMCS}}(ID_D \| R_D)$, $Cer_D = h(T_{r_D \cdot r_{MCS}}(x) \bmod p \| TS_2)$, $D_{PHI} = Cer_D \oplus PHI$ and $V_3 = h(ID_D \| Cer_D \| PHI \| R_D)$. Hence, the proposed scheme can avoid the attacker *A*'s trick for *Pat*, *Doc* and *MCS*, and resists man-in-the-middle attacks.

### 4.2.7. Resisting Stolen Verifier Attacks (Threat Model 3)

The proposed scheme does not require a verifier table maintained by *MCS* to authenticate *Pat* and *Doc*. Hence, stolen verifier attacks are not an issue.

### 4.2.8. Resisting Denial-of-Service Attacks (Threat Model 5)

In the proposed scheme, if *Pat*'s smartcard $SC : \{NID_P, Y_P\}$ is stolen or lost, a malicious attacker *A* who has $SC : \{NID_P, Y_P\}$ cannot successfully perform an undetectable online password-guessing attack to make new passwords, so the proposed scheme is resistant to denial-of-service attacks.

### 4.2.9. Compliance with HIPAA Privacy/Security Regulations

1.  Patients' understanding: The patient *Pat* has signed privacy contract $w$ during the registration phase, which clearly states how *MCS* will use, store and access *PHI*.
2.  Confidentiality (Threat model 2): This subsection concerns three phases—uploading of the patient's *PHI*, access of the patient's *PHI* and emergency-exception handling. In the uploading of the patient's *PHI*, the doctor *Doc* checks the patient's authorization through *MCS*. *Doc* and *MCS* generate the key $Cer_D$ by performing the extended chaotic Diffie–Hellman key exchange to ensure the security of *Pat*'s *PHI*. In the patient's *PHI* access phase, the doctor *Doc* checks *Pat*'s authorization $P_P$ through *MCS*. *Doc* and *MCS* generate the key $SK_D$ by performing the extended chaotic Diffie–Hellman key exchange to protect *Pat*'s *PHI*. In the emergency-exception-handling phase, the doctor *Doc* checks *Pat*'s identity through *MCS*. *Doc* and *MCS* generate the key $SK_D$ by performing the extended chaotic Diffie–Hellman key exchange to ensure the security of *Pat*'s *PHI*.
3.  Patient's control of *PHI*: Patient *Pat* generates an authorization $P_P$ and sends it to *MCS*. *MCS* checks the authorization $P_P$ that *Pat* gives *Doc*. Then, *Doc* negotiates the encryption key $SK_D$ with *MCS* and securely accesses *Pat's PHI*, which is encrypted with $SK_D$. Therefore, *Pat* must authorize access control to patient information *PHI*.
4.  *PHI* integrity (Threat model 4): The proposed scheme ensures the integrity of the medical-record information during the transmission of *PHI* by checking the confirmation message $V_4 = h(ID_D \| SK_D \| R_D \| IND_{PHI} \| PHI)$.

5.  Consent exception: When the patient has signed a privacy contract $w$ during registration and an emergency or special situation arises, $Doc$ is authorized to access the patient's medical records or health information $PHI$ from $MCS$. First, $Doc$ and $MCS$ realize mutual authentication by verifying $Cer_D$, where $Cer_D = h(T_{r_D \cdot r_{MCS}}(x) \bmod p \| TS_1)$, $r_D$ is the $Doc$'s private key and $r_{MCS}$ is the private key of $MCS$. Then, $Doc$ and $MCS$ generate the session key $SK_D$ by using the chaotic-map-based Diffie–Hellman key exchange to ensure the security of $PHI$, where $SK_D = h(T_{a \cdot b}(x) \bmod p)$. Therefore, the proposed scheme provides emergency-exception handling to protect the patients' life and rights.

### 4.3. Performance Comparison

Table 3 presents the performance of the proposed scheme and other related schemes, where $T_H$ denotes the time to execute the one-way hash function; $T_{BH}$ the time to execute a biohashing; $T_S$ denotes the time to execute symmetric en/decryption; $T_A$ denotes the time to execute asymmetric en/decryption; $T_{ECM}$ denotes the time to execute a scalar multiplication on elliptic curves; $T_{FE}$ denotes the time to execute a fuzzy extractor function; and $T_C$ denotes the time to execute extended Chebyshev chaotic maps. $T_{FE}$ and $T_{ECM}$ are considered to be the same by using the arguments presented in [35].

**Table 3.** Performance comparison.

| Phases | Registration | *PHI* Uploading | *PHI* Access | Emergency-Exception | Password-Updating |
|---|---|---|---|---|---|
| Hu et al. [3] | $6T_A + T_S$ 2.254 s | $5T_A + T_S + T_H$ 1.8874 s | $4T_A + T_S$ 1.5198 s | $2T_A$ 0.7342 s | - |
| Ray-Biswas [5] | $4T_A + T_H$ 1.4689 s | $5T_A + 2T_S + 2T_H$ 1.9393 s | $3T_A + 2T_S$ 1.2041 s | $3T_A + T_S$ 1.1527 s | - |
| Ray-Biswas [7] | $4T_A + T_H$ 1.4689 s | $3T_A + 4T_S + 3T_H$ 1.3084 s | $T_A + 4T_S + 2T_H$ 0.5737 s | $T_A + 4T_S + 2T_H$ 0.5737 s | - |
| Proposed scheme | $T_C + T_S + T_H$ 0.0688 s | $10T_C + 6T_S + 13T_H$ 0.4839 s | $10T_C + 8T_S + 14T_H$ 0.5877 s | $3T_C + 4T_S + 4T_H$ 0.2583 s | $4T_C + 3T_S + 6T_H$ 0.2248 s |
| **Phases** | **Registration** | **Authentication and Key Agreement** | | **Emergency Exception** | **Password Updating** |
| Aghili et al. [8] | $5T_H + 1T_{BH}$ 0.003 s | $28T_H + 1T_{BH}$ 0.0145 s | | - | - |
| Ali et al. [10] | $1T_{ECM} + 1T_H + 1T_{FE}$ 0.3307 s | $3T_{ECM} + 8T_H + 1T_{FE}$ 0.6644 | | - | $2T_H + 2T_{FE}$ 0.3312 |
| Fotouhi et al. [13] | $5T_H$ 0.0025 s | $34T_H$ 0.0170 s | | - | $17T_H$ 0.0085 s |
| Amintoosi et al. [15] | $5T_H$ 0.0025 s | $19T_H$ 0.0095 s | | - | $8T_H$ 0.0040 s |

Table 4 presents the hardware/software specifications and the algorithms that were used in the simulation environment, including hash function-SHA256, symmetric en/decryption algorithm-AES, asymmetric en/decryption algorithm-RSA, scalar multiplication-elliptic curve, 256-bit strings. The schemes of Hu et al. [3], Ray and Biswas [5], and Ray and Biswas [7] involve several symmetric en/decryptions and hash operations. Those schemes even require time-consuming asymmetric en/decryptions. The scheme of Ali et al. [10] also involves several scalar multiplications on elliptic curves and fuzzy extractor operations, whose computational complexities are close to that of asymmetric en/decryptions. Although Aghili et al. [8], Ali et al. [10], Fotouhi et al. [13] and Amintoosi et al. [15] provided efficient solutions for healthcare systems, their proposed schemes only consider authentication and key agreement, but their schemes do not consider PHI access and emergency exception.

**Table 4.** Simulation environment.

| Hardware/Software Specification |
| --- |
| Intel Xeon CPU E3-1231 v3 3.4 GHz |
| 8 G Memory |
| Windows Server 2008 |
| Visual Studio 2012 and C++ programming language |
| Input string length- 256 bits |
| **Used Algorithms** |
| Hash function: SHA256 |
| Symmetric en/decryption algorithm: AES |
| Asymmetric en/decryption algorithm: RSA |
| Scalar multiplication: Elliptic curve |
| Extended Chebyshev chaotic maps |

Figure 7 illustrates the response-time statistics of the proposed scheme and other related schemes, which can fully provide the registration phase, PHI uploading phase, PHI access and emergency-exception-handling phase and include the schemes of Hu et al. [3], Ray and Biswas [5] and Ray and Biswas [7]. Compared with other related schemes, the proposed scheme is much faster than in the registration phase; it increases computing performance by at least 63.0% in the PHI uploading phase, and increases computing performance by at least 54.9% in the emergency-exception-handling phase. The proposed scheme completely considers PHI uploading, access and emergency exceptions, and employs only extended chaotic maps, symmetric en/decryptions and hash operations, which have low computation burdens. Therefore, the proposed scheme not only provides more functionality, but also retains efficiency in computations.

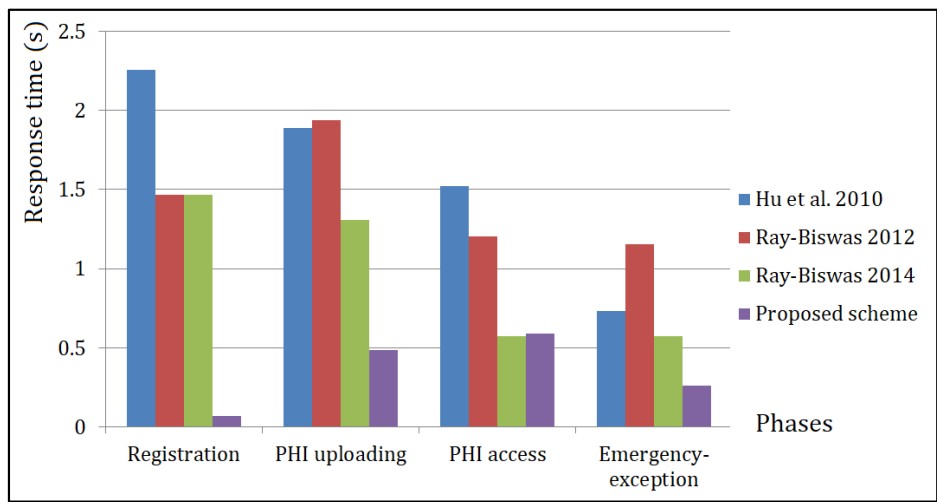

**Figure 7.** The response time of other related schemes in terms of registration phase, PHI uploading phase, PHI access and emergency-exception-handling phase.

Table 5 compares the communication of the proposed scheme and other related schemes in term of required messages. The schemes of Hu et al. [3], Ray and Biswas [5] and Ray and Biswas [7] require more messages than the proposed scheme in the PHI uploading and PHI access phases. Furthermore, the proposed scheme has similar numbers of communication messages with those of Aghili et al. [8], Ali et al. [10], Fotouhi et al. [13] and Amintoosi et al. [15], and comprehensively considers PHI uploading, PHI access and emergency-exception phases.

**Table 5.** Communication comparison.

| Phases | *PHI* Uploading | *PHI* Access | Emergency-Exception |
|---|---|---|---|
| Hu et al. [3] | 5 | 5 | 2 |
| Ray and Biswas [5] | 5 | 4 | 4 |
| Ray and Biswas [7] | 5 | 5 | 2 |
| Proposed scheme | 4 | 4 | 2 |
| Aghili et al. [8] | 4 | | - |
| Ali et al. [10] | 3 | | - |
| Fotouhi et al. [13] | 4 | | - |
| Amintoosi et al. [15] | 4 | | - |

### 4.4. Functionality Comparison

Table 6 compares and the proposed scheme and related schemes in terms of functionality, and specifically the meeting of security requirements and resistance of potential attacks. The schemes of Hu et al. [3], Ray and Biswas [5] and Ray and Biswas [7] are developed by using RSA and the scheme of Ali et al. [10] are developed by using scalar multiplications on elliptic curves, thus requiring time-consuming computations. The schemes of Ray and Biswas [5], Ray and Biswas [7] and Ali et al. [10] perform verification processes by using certificates. The schemes of Hu et al. [3], Aghili et al. [8], Fotouhi et al. [13] and Amintoosi et al. [15] perform verification processes by using smartcards. The schemes of Aghili et al. [8], Fotouhi et al. [13] and Amintoosi et al. [15] are developed by using hash operations. The proposed scheme is developed by using hash operations and chaotic maps, which are lightweight operations. Although the schemes of Aghili et al. [8], Fotouhi et al. [13] and Amintoosi et al. [15] have higher efficiency in computation, their schemes do not consider PHI access and emergency exceptions, and thus cannot provide complete security requirements. Additionally, only the proposed scheme completely considers PHI uploading, access and emergency exceptions; provides completed security requirements, including updated passwords, patients' authorization and patients' control; and resists potential attacks, including password-guessing attacks, impersonation attacks, replay attacks and stolen verifier attacks. Accordingly, the proposed scheme is efficient and exhibits greater functionality than the other schemes.

**Table 6.** Functionality comparison.

| Scheme | Hu et al. [3] | Ray and Biswas [5] | Ray and Biswas [7] | Aghili et al. [8] | Ali et al. [10] | Fotouhi et al. [13] | Amintoosi et al. [15] | Proposed Scheme |
|---|---|---|---|---|---|---|---|---|
| Used Algorithm | RSA | RSA | RSA/AES | Hash | ECC | Hash | Hash | ECM |
| User Verification | SC | PKC | PKC | SC | PKC | SC | SC | PKC |
| Providing MA | YES | NO | YES | YES | YES | YES | YES | YES |
| Providing UP | NO | NA | NA | NO | YES | YES | YES | YES |
| Providing PA | NO | NO | NO | NO | NO | NO | NO | YES |
| Providing PC | NO | NO | NO | NO | NO | NO | NO | YES |
| Resisting PGA | NA | NA | NA | YES | YES | YES | YES | YES |
| Resisting IA | NO | NO | YES | NO | YES | YES | YES | YES |
| Resisting RA | NO | NO | YES | YES | YES | YES | YES | YES |
| Resisting MMA | YES | NO | YES | YES | YES | YES | YES | YES |
| Resisting SVA | NO | NA | NO | YES | YES | NO | YES | YES |

SC: smart card; PKC: public-key certificate; MA: mutual authentication; UP: updated password; PA: patients' authorization; PC: patients' control; PGA: password-guessing attacks; IA: impersonation attacks; RA: replay attacks; MMA: man-in-the-middle attacks; SVA: stolen verifier attacks; ECM: extended chaotic maps.

## 5. Conclusions

This work develops a user-authentication and key-agreement scheme that exploits the favorable characteristics and speed of Chebyshev polynomials to provide high computational efficiency and comply with HIPAA privacy/security regulations. The proposed scheme solves security problems that afflict related schemes, such as the accessing of patient information without patient authorization, the inability to perform multiple verifications simultaneously, and others. This proposed key-agreement scheme depends on a certificate-management center to enable doctors, patients and authentication servers to realize mutual authentication through certificates and thereby reduce the number of rounds of communications that are required. The proposed scheme provides all of the security functions of related schemes, while overcoming their limitations and offering greater efficiency. It is more secure and compliant with HIPAA privacy/security regulations, so it is more suitable for real-world environments.

Since the proposed scheme needs to comply with privacy/security regulations and needs to consider more contexts, it is more complicated than other schemes applied to healthcare systems. Future work plans to simplify the process of the proposed scheme, comply with the general principles of privacy/security regulations and be applicable to practical application scenarios.

**Author Contributions:** Conceptualization, Y.-P.H. and T.-F.L.; methodology, T.-F.L. and G.-J.S.; validation, Y.-P.H. and K.-C.L.; formal analysis, T.-F.L. and K.-C.L.; investigation, Y.-P.H., T.-F.L., K.-C.L. and G.-J.S.; writing—original draft preparation, K.-C.L. and G.-J.S.; writing—review and editing, Y.-P.H. and T.-F.L.; supervision, Y.-P.H. and T.-F.L.; project administration, T.-F.L.; funding acquisition, T.-F.L. All authors have read and agreed to the published version of the manuscript.

**Funding:** This research was funded by the Ministry of Science and Technology of the Republic of China, grant number MOST 109-2221-E-320-003 and MOST 110-2221-E-320-005-MY2.

**Institutional Review Board Statement:** Not applicable.

**Informed Consent Statement:** Not applicable.

**Acknowledgments:** Ted Knoy is appreciated for his editorial assistance.

**Conflicts of Interest:** The authors declare no conflict of interest.

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
