# Peer review of "Extended Chaotic-Map-Based User Authentication and Key Agreement for HIPAA Privacy/Security Regulations"

_applsci, doi:10.3390/app12115701_

Round 1

Reviewer 1 Report

This paper presents a protcol for security medical sharing using chaotic maps. 

I found several concerns with the paper mainly with the presentation, while the technical aspects are very hard to assess with so many tangling acronyms, symbols and equations presented one after the other. 

In section 1.5 what do you mean by "The proposed scheme solves the security problems of previous schemes"? What are the security issues of previous works and how do you solve them? You have carried out a performance comparison not a security comparison. 

Also in section 1.5 what is the difference between contibution 1 and 2. It seems that these are  the same. 

It is strange that the authors did not use BAN login or AVISPA or ProVerif to prove formally their security claims. Without formal analysis it is very difficiult to assess the technical parts of the protocol. Besides, the protocol sseems to be too complex and lengthy, and the authors do not discuss at all the performance penalties of communciation overheads.

The authors should address the above comments. 

Additinally,

Some papers are also missing in the comparison such as:

Ryu, Jihyeon, Dongwoo Kang, and Dongho Won. "Improved Secure and Efficient Chebyshev Chaotic Map-Based User Authentication Scheme." IEEE Access 10 (2022): 15891-15910.

Zhai, Xiao-Ying, and Jian Wang. "A multi-server biometric authentication scheme based on extended chaotic map for telecare medical information system." Multimedia Tools and Applications (2022): 1-21.

Author Response

Q1. I found several concerns with the paper mainly with the presentation, while the technical aspects are very hard to assess with so many tangling acronyms, symbols and equations presented one after the other.

Ans.: We thank the reviewer for this constructive suggestion. Indeed, the proposed scheme involves more actor roles in the system model and provides more functionality.  Requiring more tangled acronyms, symbols and equations complicates our scheme. We have them as simply as possible.

Q2. In section 1.5 what do you mean by "The proposed scheme solves the security problems of previous schemes"? What are the security issues of previous works and how do you solve them? You have carried out a performance comparison not a security comparison.

Ans.: We thank the reviewer for this constructive suggestion. The proposed scheme completely considers PHI uploading, access and emergency exception, provides completed security requirements, including updated password, patients’ authorization, patients’ control and resists potential attacks, including password guessing attacks, impersonate attacks, replay attacks and stolen verifier attacks. We have revised the descriptions according to the reviewer’s comments. The descriptions are shown on Page 4, Sec. 1.5 Lines 170-173; and on Page 21, Sec. 4.4, Lines 709-713.

Q3. Also in section 1.5 what is the difference between contribution 1 and 2. It seems that these are the same.

Ans.: We thank the reviewer for this constructive suggestion. We have revised the contributions according to the reviewer’s comments. The descriptions are shown on Page 4, Sec. 1.5, Lines 168-173.

Q4. It is strange that the authors did not use BAN login or AVISPA or ProVerif to prove formally their security claims. Without formal analysis it is very difficult to assess the technical parts of the protocol. Besides, the protocol seems to be too complex and lengthy, and the authors do not discuss at all the performance penalties of communication overheads.

Ans.: We thank the reviewer for this constructive suggestion. We have included the authentication proof using BAN logic and communication comparison and revised the descriptions according to the reviewer’s comments. The descriptions are shown on Pages 13-16, Sec.4.1 and on Page 20, Sec. 4.3, Lines 685-693 and Table 4.

Q5. Some papers are also missing in the comparison such as:

Ryu, Jihyeon, Dongwoo Kang, and Dongho Won. "Improved Secure and Efficient Chebyshev Chaotic Map-Based User Authentication Scheme." IEEE Access 10 (2022): 15891-15910.

Zhai, Xiao-Ying, and Jian Wang. "A multi-server biometric authentication scheme based on extended chaotic map for telecare medical information system." Multimedia Tools and Applications (2022): 1-21.

Ans.: We thank the reviewer for this constructive suggestion. We have included some suggested important references and revised the descriptions according to the reviewer’s comments. Since these schemes are client/ server-based, the descriptions are included in Sec. 1.4. The descriptions are shown on Pages 3-4, Sec. 1.4, Lines 140-146.

Reviewer 2 Report

 In this article, authors proposed key agreement scheme which depends upon a certificate management center to enable doctors, patients and authentication servers to realize mutual authentication through certificates and thereby reduce the number of rounds of communications. The proposed scheme provides all of the security functions of related schemes. 

Authors further discussed a future work plans to simplify the process of the proposed scheme and comply with the general principles of privacy/security regulations, and be applicable to practical application scenarios.

In all of the above changes, I am satisfied with revision, but authors must add the following reference and cite properly

1)Strong Convergence Theorems for a Finite Family of Enriched Strictly Pseudocontractive Mappings and ΦT-Enriched Lipschitizian Mappings Using a New Modified Mixed-Type Ishikawa Iteration Scheme with Error, 2)Interpolative C´iric´-Reich-Rus-type best proximity point results with applications

Conclusion:
The Paper is worthy, the results seem correct, and the article can be published after this minor revision.

Author Response

Reviewer 2:

Comments and Suggestions for Authors

 In this article, authors proposed key agreement scheme which depends upon a certificate management center to enable doctors, patients and authentication servers to realize mutual authentication through certificates and thereby reduce the number of rounds of communications. The proposed scheme provides all of the security functions of related schemes.

Authors further discussed a future work plans to simplify the process of the proposed scheme and comply with the general principles of privacy/security regulations, and be applicable to practical application scenarios.

In all of the above changes, I am satisfied with revision, but authors must add the following reference and cite properly

1)Strong Convergence Theorems for a Finite Family of Enriched Strictly Pseudocontractive Mappings and ΦT-Enriched Lipschitizian Mappings Using a New Modified Mixed-Type Ishikawa Iteration Scheme with Error, 2)Interpolative C´iric´-Reich-Rus-type best proximity point results with applications

Conclusion:

The Paper is worthy, the results seem correct, and the article can be published after this minor revision.

Ans.: We thank the reviewer for this constructive suggestion. We have included some suggested important references and revised the descriptions according to the reviewer’s comments. The descriptions are shown on Page 4, Sec. 1.5

Reviewer 3 Report

The authors have addressed the comments in a satisfactory way

Author Response

The authors have addressed the comments in a satisfactory way

Ans.: We thank the reviewer for this constructive suggestion.

Round 2

Reviewer 1 Report

The paper has been improved